# Superelastic graphene aerogel-based metamaterials

**Mingmao Wu[1,2], Hongya Geng[3], Yajie Hu [2], Hongyun Ma [2], Ce Yang[4], Hongwu Chen [2], Yeye Wen [2], Huhu Cheng[2], Chun Li [2], Feng Liu [5], Lan Jiang [6] & Liangti Qu [2,4] ✉**

Ultralight, ultrastrong, and supertough graphene aerogel metamaterials combining with multi-functionalities are promising for future military and domestic applications. However, the unsatisfactory mechanical performances and lack of the multiscale structural regulation still impede the development of graphene aerogels. Herein, we demonstrate a laser-engraving strategy toward graphene meta-aerogels (GmAs) with unusual characters. As the prerequisite, the nanofiber-reinforced networks convert the graphene walls' deformation from the microscopic buckling to the bulk deformation during the compression process, ensuring the highly elastic, robust, and stiff nature. Accordingly, laser-engraving enables arbitrary regulation on the macro-configurations of GmAs with rich geometries and appealing characteristics such as large stretchability of 5400% reversible elongation, ultralight specific weight as small as $0.1 \, mg \, cm^{-3}$, and ultrawide Poisson's ratio range from −0.95 to 1.64. Additionally, incorporating specific components into the pre-designed meta-structures could further achieve diversified functionalities.

Graphene aerogels (GAs) integrating ultralight, unexpected mechanical strength, and unprecedented fatigue resistance are promising "3U" mechanical materials fitting harsh environments in aerospace, defense, and energy-related fields[1–4]. However, pursuing the unique "3U" properties for the materials community is still fraught with challenges due to mutually exclusive qualities between strength and resilience at low density[5]. Additionally, the lack of multiscale structural regulation from tailorable macro to well-ordered micro configurations also restricts GAs applications in multi-functional scenarios such as wearable electrical circuits, soft robots, and stimulus-responsive devices[6,7]. Therefore, building strong and elastic GAs with controllable macro and microarchitectures, as well as understanding the mechanics that drives GAs' superior mechanical properties, are fascinating.

On the shaping of macro-morphology, freeze-drying is a widely used method to obtain GAs with the original appearance of molds[6], which can to some extent regulate the arrangements of graphene sheets, rendering cellular, porous, and hyperbolic networks[8–12]. However, it faces the difficulty to obtain precise and specific shapes with high performances matching the variable application scenario[6]. Three-dimensional (3D) printing is also a promising extrusion technique to fabricate lattice and periodic GAs with customizable structures[13,14]. Nevertheless, the need to use a high concentration of graphene oxide (GO) dispersion for sufficient viscosity becomes a restrictive condition to downsize the printed structures[15]. The high viscosity makes GO strenuous to adjust the microstructure, generally leading to a purely stochastic network of GAs with weak sheet-to-sheet interactions and unsatisfactory mechanical behaviors[13,14].

[1]Key Laboratory of Eco-materials Advanced Technology, College of Materials Science and Engineering, Fuzhou University, 350108 Fuzhou, China. [2]Key Laboratory of Organic Optoelectronics & Molecular Engineering, Ministry of Education, Department of Chemistry, Tsinghua University, 100084 Beijing, P. R. China. [3]Department of Materials Imperial College London Prince Consort Road, London SW7 2AZ, UK. [4]State Key Laboratory of Tribology, Department of Mechanical Engineering, Tsinghua University, 100084 Beijing, P. R. China. [5]State Key Laboratory of Nonlinear Mechanics, Institute of Mechanics, Chinese Academy of Sciences, 100190 Beijing, China. [6]Laser Micro-/Nano-Fabrication Laboratory, School of Mechanical Engineering, Beijing Institute of Technology, 100081 Beijing, P. R. China. ✉e-mail: lqu@mail.tsinghua.edu.cn

On the other hand, the interior microstructure of GAs has a strong influence on their macro-mechanical properties[10,16,17]. Belonging to the open cell model, the deformation of GAs follows three procedures in the sequence of linear elasticity, collapse, and densification throughout the compression process[18–21]. The failure of GAs is accompanied by the formation of microscopic hinges, which occur when the graphene walls approach their maximum bending moment and then collapse. Meanwhile, the elastic moduli of GAs are also associated primarily with the bending stiffness of graphene walls[20]. Hence, to attain high stiffness and elasticity of the GAs, efforts have been made to improve the bending stiffness of graphene walls through hydrothermal treatment, molecule crosslinking, and polymer reinforcement[22–25]. However, the specific elastic bending stiffness of a sheet-like structure is intrinsically inferior to that of tubular and fibrillar materials[8,16,26,27].

Herein, we present a laser-engraving strategy towards graphene meta-aerogels (GmAs) possessing multi-functional macroscopic structures and highly ordered micro-networks. GmAs can be rapidly and precisely engineered into any configurations including line, plane, 3D lattices, and hole bulks, giving record high features such as super stretchability (5400% elongation), low specific weight (0.1 mg cm$^{-3}$), and ultrawide Poisson's ratio ($\nu$) range ($-0.95 < \nu < 1.64$). The microstructures of GmAs are composed of well interlocked sub-micron fibers and graphene sheets. Benefiting from the 1D nanofiber-reinforced 2D sheet structure, the bending stiffness of graphene walls increases significantly. Thus, during the compression process, the internal deformation mode of GmAs is a stable bulk deformation rather than the microscopic buckling for soft GAs, which results in the good mechanical properties of GmAs including good robustness, high

strength and stiffness, and full shape recovery after arbitrary compression, outperforming most carbon aerogels. Furthermore, the simple manufacturing process allows the incorporation of other functional components into the ordered structure, enabling the production of objects with pre-defined geometry and functions. As prototypes, ceramic aerogels and magnetically responsive aerogels were prepared, indicating great potential for multi-functional applications.

## Results

### Macro-structural regulation of GAs

Traditional engraving has always been an important way for human beings to sculpture objects with unique functions and rich values, such as creating labor-saving tools and gorgeous artwork (Fig. 1a). Current micro-nano structure regulation is facing a higher level of precision requirements. Especially for the GA, its soft and weak attributes make it difficult to manipulate its shape. Laser-engraving refers to utilizing the micron-sized laser beam, which can induce the local thermal effect to break the covalent bonds, thereby accurately cutting and modifying the micro or nanomaterials, even if it is a soft material[28,29]. Under the digital laser engraving on the target surfaces of GAs, specific structures unavailable previously can be obtained facily (Fig. 1b), e.g., the peony flower structures (Fig. 1c). Moreover, the pieces can be patterned with well-designed shapes and rabbets, which can even be assembled into a stereoscopic eagle (Fig. 1d), demonstrating immense possibilities of laser-induced morphology design. Furthermore, when combined with the mechanical structure design[30], the serpentine structure, re-entrant structure, and spiral structure, can be tailored on-demand, endowing them with 1200%, 133%, and 5400% elongation, respectively (Fig. 1e–g).

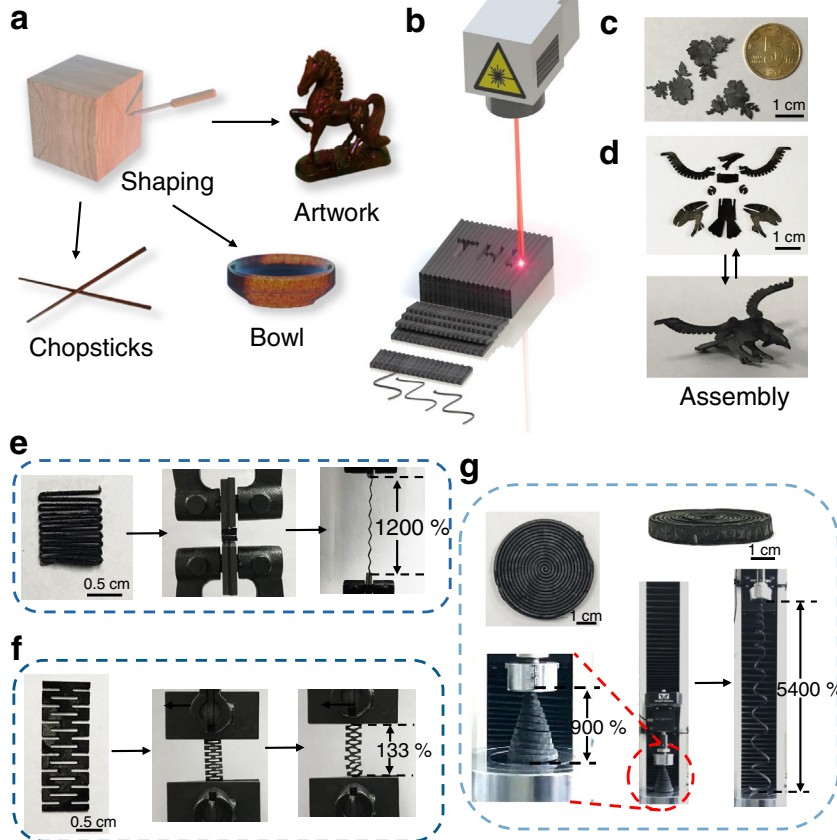

**Fig. 1 | Traditional engraving and laser-engraving for graphene aerogels (GAs) with arbitrary geometries. a** Schematic illustration of traditional engraving for tools and artwork, such as a pair of chopsticks, a bowl, and a sculpture of steed. **b** Schematic illustration of laser-engraving on GA with arbitrary shapes. **c** The peony flower pattern of GAs with fine structures as small as a fifty cents coin after laser-engraving. **d** The pieces of GAs with elaborate shapes and rabbets can be reversibly assembled into a stereoscopic eagle. **e** The GAs with serpentine structure, stretched reversibly with 1200% tensile strain. **f** The GAs with re-entrant structure, showing reversible strains of 133%. **g** The spiral GAs stretched reversibly with 900% and even up to 5400% tensile strain.

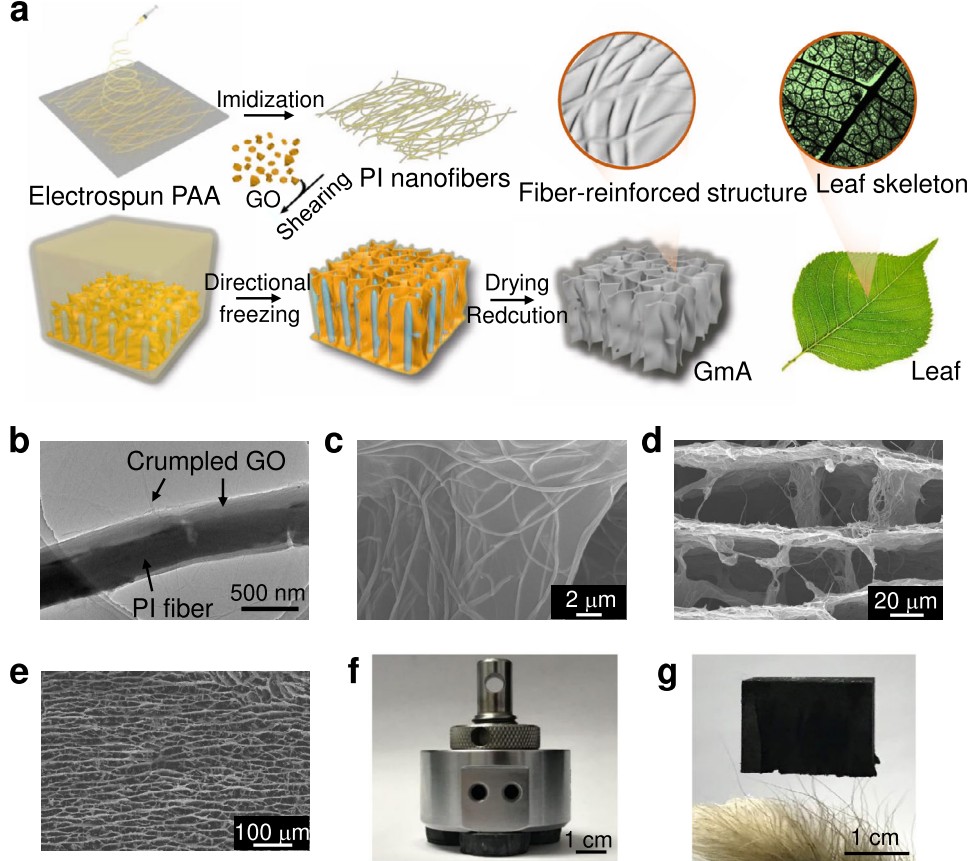

**Fig. 2 | Fabrication and structure of graphene meta-aerogels (GmAs).**
**a** Schematic illustrations of fabrication procedures of the GmAs and a leaf with the optical photograph of the leaf skeleton. **b** TEM image of the graphene oxide (GO) sheets covering polyimide (PI) nanofibers. **c**–**e** The cross-section SEM images of GmAs at different scales. **f** Three GmAs support weight over 6000 times their weight without any macroscopic deformation. **g** A GmA (2 cm × 2 cm × 1.8 cm) stands on Setaria viridis.

It is difficult to achieve for pristine GAs due to the brittleness of the graphene sheets[31]. GAs are generally soft, making them unable to tolerate large deformations while maintaining their functional shapes. Constructing the GAs with excellent mechanical performances is the critical prerequisite to fulfill the excellent characteristics and morphologic design.

## Design principal and micro-structural regulation of GmAs

To enhance mechanical performances of GA, one dimensional (1D) nanofibers were introduced to GA framework. Polyimide (PI) nanofibers were optimally selected because of their rigid aromatic chain structure and good processability, advantageous to firmly bind all the graphene sheets through the π−π interaction (Supplementary Figs. 1, 2)[32]. Fig. 2a illustrates the fabrication procedures of GmAs. The PI mat was prepared by electrospinning and thermal imidization[33]. And followed by a high-speed shearing, the cutting-off PI nanofibers with an average diameter of 207 nm and an average length of 88 μm were prepared (Supplementary Fig. 3), which were favorably encapsulated by GO sheets (Fig. 2b). The mixture of PI nanofibers with GO was freeze-dried, followed by annealing reduction treatment to obtain conductive GmAs (Supplementary Fig. 4). The cross-section SEM images display the hierarchical microarchitectures of GmAs at distinct scales: (i) on the surface of graphene, nanofibers cover on the graphene sheets and entangle with each other (Fig. 2c). Just like a leaf skeleton (Fig. 2a and Supplementary Fig. 5), it is the naturally optimized strategy for structural strengthening at the minimum cost, which can support the shape of the main body even under severe conditions. (ii) in each cellular unit, nanofibers firmly bind the whole graphene sheets forming a continuous network (Fig. 2d), and (iii) at

the outline, the GmA exhibits a long-range aligned lamella architecture up to several centimeters (Fig. 2e and Supplementary Fig. 6). Accordingly, these multiscale configurations, which include interlocking 1D nanofiber and 2D sheets building blocks, continuously binding network units, and highly ordered graphene wall frameworks, provide GmAs with solid appearances in the case of nearly 100% porosity, while allowing them to support a weight of 6000 times their own weight without any macroscopic deformation (Fig. 2f), and possess featherlight density (3 mg cm$^{-3}$), stably standing on slimsy Setaria viridis (Fig. 2g).

## Mechanical performances of GmAs

To investigate the effects of 1D nanofiber-reinforced 2D graphene sheet structure on the mechanical performances, the compressive experiments were conducted on the as-prepared GAs (Supplementary Fig. 7), including the GmA and the pristine GA (PGA). Figure 3a exhibits the high elasticity of the GmA under uniaxial plane compression at increasing strain (ε). Even with a loading ε up to 90%, the unloading ε of GmA can return 0%, proving its full recovery capabilities under large deformation (Supplementary Fig. 8). The cycling loading-unloading tests reveal the fatigue resistance of GmA, which can maintain 100% ε retention and over 95% stress (σ) retention after 1000 cycles of 50% compression ε (Fig. 3b and Supplementary Fig. 7b). The long-term cycling with 82% stress retention at a large compressive ε of 80% further confirms the durability of GmA throughout the wide compression range (Supplementary Fig. 9). Furthermore, after pre-compression treatment, the stress retention of GmA can be significantly enhanced to 99% and 90% for the 50% and 80% compression ε, respectively. These performances outperform most carbon-based aerogels at similar

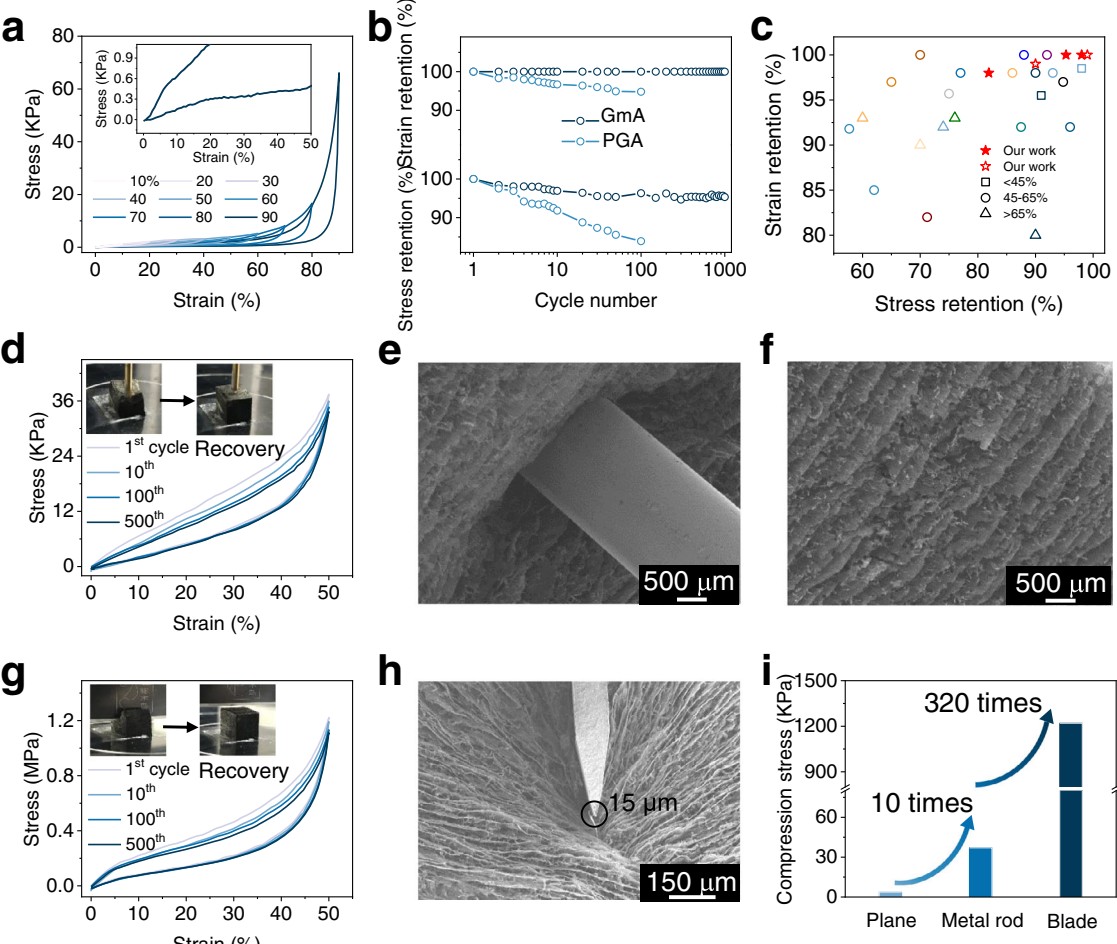

**Fig. 3 | Superior mechanical performances of GmAs. a** Compressive stress–strain curves of GmA at strain up to 90%, inset the zoom-up curve of 90% strain. **b** The strain and stress retention of GmA and pristine graphene aerogel (PGA) during 1000 cycles at 50% strain. **c** Extensive comparison of the strain retention and stress retention of GmA and reported carbon based aerogels[8,9,11,14,23–26,34–43]. Stars represent the results in this work, and the open stars represent the GmA after pre-compression. **d** Stress–strain curves of GmA at 50% strain for 500 cycles under metal rod compression with a compressive area of only 3.1 mm². The insets are the photographs of GmA under compression and after recovery. **e** SEM image of the GmA under the metal rod compression. **f** SEM image of GmA after release of the metal rod compression. **g** Stress–strain curves of GmA at 50% strain for 500 cycles under sharp blade compression with a narrow compressive area of 0.12 mm². The insets are the photographs of GmA under compression and after recovery. **h** SEM image of the GmA under blade cutting. The black circle marks the blade edge width is only 15 µm. **i** Histogram of compression stress for different compressive mode at 50% strain. Source data are provided as a Source Data file.

compressive $\varepsilon$ (Fig. 3c and Supplementary Table 1)[8,9,11,14,23–26,34–43], indicating the good resilience and structural robustness of GmA. In contrast, PGA exhibits a severe collapse of 16% stress loss with an irreversible deformation over 5% after only 100 cycles of compression with 50% $\varepsilon$ (Fig. 3b and Supplementary Fig. 7a).

GmAs with long-range continuous frameworks can withstand severe compression deformation. Figure 3d–h displays that the GmA bears the repeated compression applied by a metal rod and a blade, which compress the GmAs with the small area of 3.1 mm² and 0.12 mm². Especially for the blade cutting, the sharp edge is only 15 µm. At these harsh compression conditions, the GmA can still maintain its original shape and well survive after 500 cycles at 50% $\varepsilon$ (Fig. 3d, g) and even 80% $\varepsilon$ (Supplementary Fig. 10), which benefits from the connected graphene walls weaved by 1D nanofibers that can efficiently dissipate the locally concentrated energy (Fig. 3e, f, h). Notably, the stresses at 50% $\varepsilon$ of the rod compression and blade cutting significantly increase to 37 KPa and 1.2 MPa, which are 10 times and 320 times higher than that of the plane compression (Fig. 3i), demonstrating the structural stiffness and good tolerance of GmA. Moreover, the GmA can withstand stretching with a breaking elongation of up to 6% (Supplementary Fig. 11), which offers the GmA with more deformation possibilities

such as bending and folding behaviors (Supplementary Fig. 12). Additionally, Ashby charts of strength and modulus versus density for various types of materials were depicted in Supplementary Fig. 13a, b, which further confirm the unique properties of ultralight GmA. Considering the great tolerance and elasticity, GmAs are promising for further processability (Supplementary Fig. 13c).

## Mechanism investigation of mechanical properties

The compression and release of GmA and PGA were in-situ monitored to track the deformation processes. As displayed in Fig. 4 and Supplementary Fig. 14, the PGA possesses the similar parallel structure to the GmA. During the loading–unloading process, the graphene walls in PGA firstly undergo local instability and microscopic buckling (arrows in Fig. 4b), subsequently sharp fold (arrows in Fig. 4c), as well as fractures at large deformation (circle in Fig. 4a–d). The unsatisfying compression behavior is mainly due to the poor connectivity of graphene sheets, which can only bend solely in the absence of the support of the whole frameworks. When the graphene walls exceed their yield stress, they will be microscopic buckled permanently (Fig. 4a and Supplementary Fig. 14b)[20]. For the GmA, its 1D nanofibers reinforced 2D sheet structure binds the whole graphene sheets into one

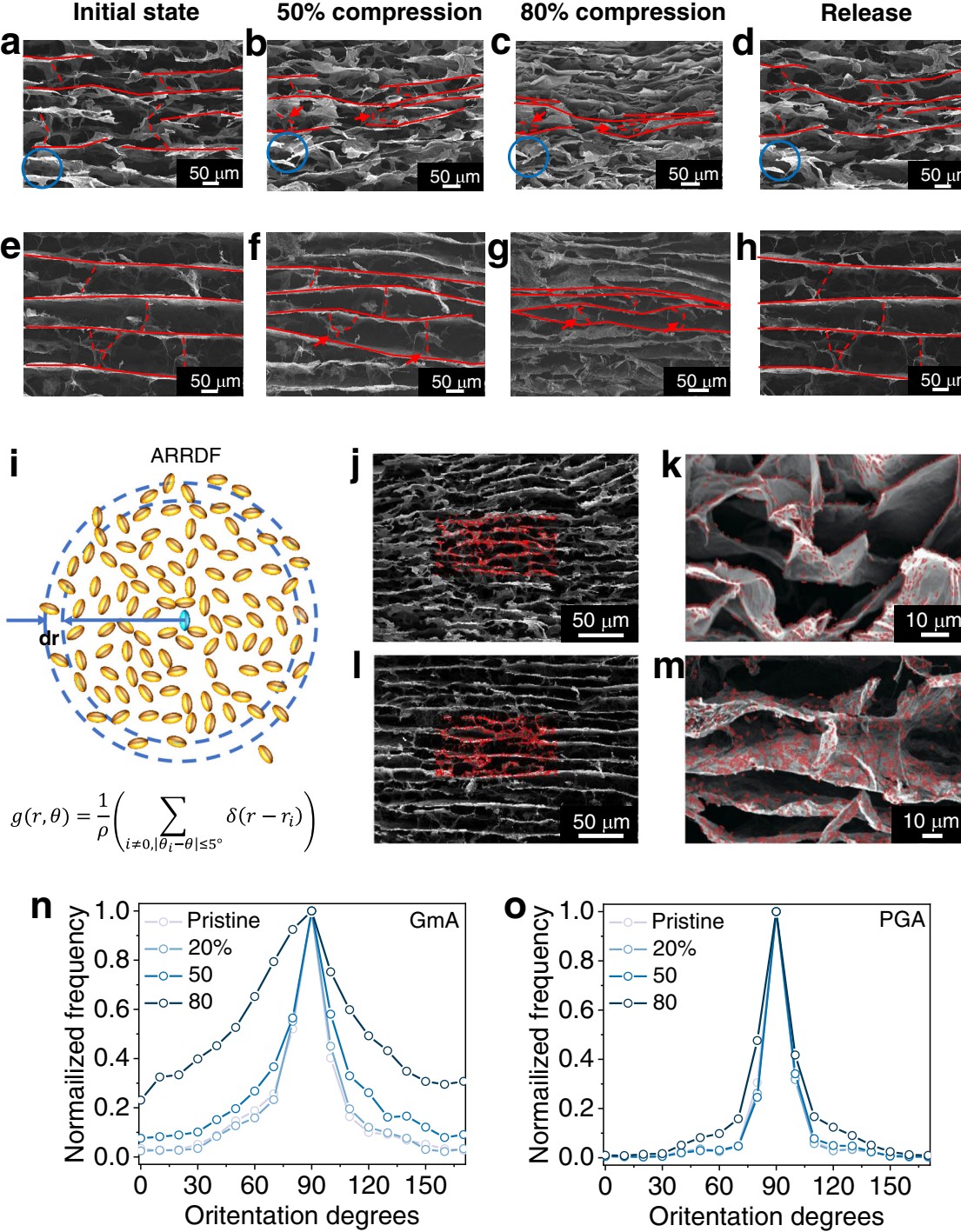

**Fig. 4 | Strengthen mechanism analysis of SEM observation and ARRDF.**
**a**–**d** SEM images of PGA under different compression state, **a** 0%, **b** 50%, **c** 80%, **d** release. Arrows in **b** mark the local instability parts. Arrows in **c** mark the extreme bending parts. Circles points out the fracture process of the graphene wall. **e**–**h** SEM images of GmA under different compression state, **e** 0%, **f** 50%, **g** 80%, **h** release. Arrows in **f** and **g** mark the stable bulk deformation process. **i** The scheme and functions of angle-resolved radial distribution function (ARRDF) describe how density varies as a function of distance from a reference particle (blue one). **j**–**m** SEM images of PGA and GmA under compression conditions with corresponding texture recognition ellipses, **j** PGA, **l** GmA, part of which are zoomed in **k** and **m**. **n**, **o** The normalized frequency of $g(r, \theta)$ at the pixel distance of 500 extracted to show the relative distribution, **n** GmA, **o** PGA. Source data are provided as a Source Data file.

entirety (Fig. 4e–h and Supplementary Figs. 6 and 14c). During the compression process, this strong framework prefers to a stable bulk deformation (arrows in Fig. 4f, g) instead of the local bending, which can resist the stress concentration, thus protecting the graphene walls from buckling failure, even for the bridge skeletons between the lateral lamellas (Supplementary Fig. 14d). Therefore, the GmA can

fully spring back to its original state after load releasing (Supplementary Fig. 14c).

Herein, to further confirm the difference of the deformation process rather than personal inclination in visual observation, an angle-resolved radial distribution function (ARRDF, $g(r, \theta)$) is established and directly extracted from the SEM images to statistically

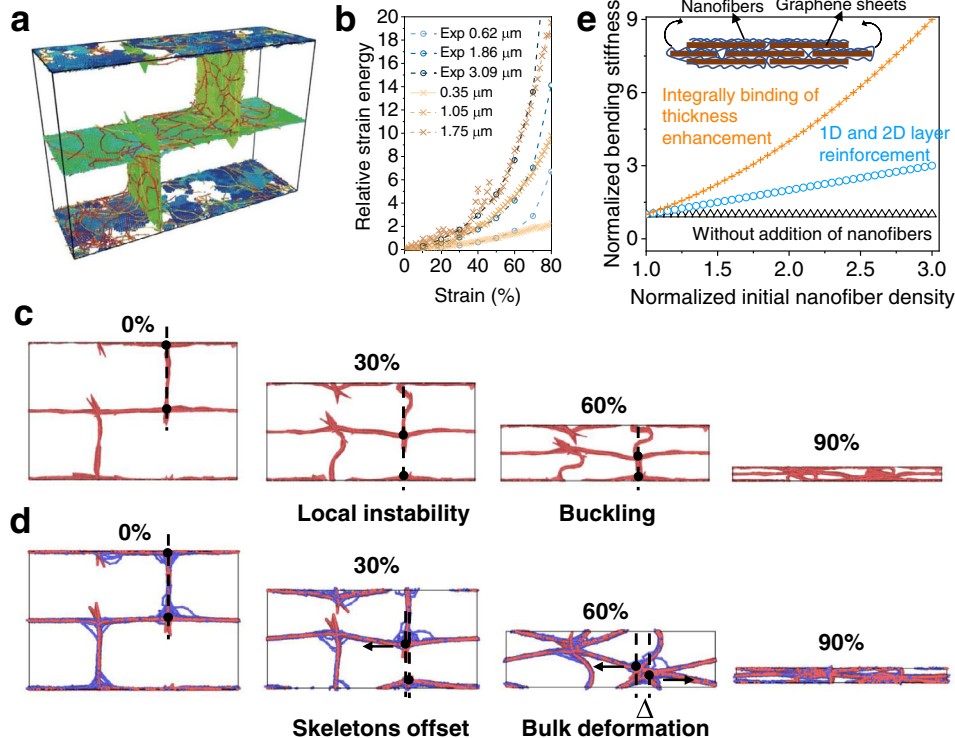

**Fig. 5 | Molecular dynamic simulations of strengthen mechanism. a** Calculation model, where different graphene sheets and fibers are marked in different colors. **b** Relative strain energy versus strain, where three experimental samples with typical thicknesses of 0.62, 1.86 and 3.09 μm, are compared with the simulation results with their skeleton thicknesses (0.35, 1.05, 1.75 μm). For experimental data, the strain energy under 50% compression with typical thicknesses of 0.62 μm is chosen to normalize strain energy, and correspondingly the strain energy under 50% compression with skeleton thickness of 0.35 μm for simulation results. **c** The simulation deformation process of PGA. **d** The simulation deformation process of GmA. The Δ is the offset of the horizontal distance between the junction of the lateral skeletons and bridge skeleton under compression state. **e** Initial nanofiber density dependence of bending stiffness of graphene skeleton. Blue circle marks the growth rate where only one dimensional (1D) and two dimensional (2D) layer reinforcement is considered, while orange line with '+' marks the additional enhancement from thickness is included. The black triangle marks the PGA without nanofibers. The schematic diagram of the 1D nanofibers reinforced 2D structure is shown in the inset. Source data are provided as a Source Data file.

depict orientation frequency of graphene walls (Fig. 4i and the details in Supplementary note 1 and Fig. 15).

$$g(r,\theta) = \frac{1}{\rho} \left( \sum_{i \neq 0, |\theta_i - \theta| \leq 5^\circ} \delta(r - r_i) \right) \quad (1)$$

where $\rho$ is the density and $r$ is the radius. In the SEM images of the compression process, each piece of texture is recognized by an ellipse according to the orientation of the texture (Fig. 4j–m and Supplementary Fig. 16). Taking the vertical axis as the reference orientation of ellipse (Supplementary Fig. 17), the orientation variational tendencies of all the ellipses can be statistically recorded by the $g(r, \theta)$, which can further reflect the variational trend of the bending angle of graphene walls during the compression process. Specifically, under the initial state, the statistical curves for GmA and PGA exhibit the similar mountain-like shapes with the highest frequency at 90° orientation, representing the most frequent horizontal graphene walls (Fig. 4n, o and Supplementary Figs. 18, 19). As the compression progressing to 20%, 50%, and 80%, the curves for PGA are still almost unchanged. While in GmA, the frequency of other orientation relative to 90° increases significantly, suggesting the bridged graphene walls between the parallel lamellas prefer tilt rather than the severe folding in PGA during the compression process. Thus, a tilt-induced bulk deformation instead of microscopic buckling takes place with 1D nanofibers reinforced 2D structure.

Molecular dynamic simulations validate the strengthen mechanism of deformation transformation derived from the increasing of bending modulus. A GmA model composing of 2D graphene sheets

and 1D nanofibers was constructed according to experimental observations (model details in Supplementary note 2 and Fig. 5a). In order to verify the validity of the established model, a series of GmAs with different graphene wall thicknesses of 0.62, 1.86 and 3.09 μm were experimentally prepared and their compressive performances were recorded (Supplementary Figs. 20, 21). Meanwhile, the GmAs models with similar thickness increase trends of 0.35, 1.05, 1.75 μm (about 1: 3: 5) were simulated. After normalizing the experiment and simulation results, the experimental relative strain energy curves nearly overlap with the simulation results, indicating this model could well describe the inner microstructures of GmA (Fig. 5b). Moreover, on the basis of theoretical model, the bulk deformation process can also be observed (Fig. 5c, d), being consistent with the SEM observation and ARRDF statistics. For convenient observation, the transverse offset (Δ) of bridge trusses was marked. During the process of gradual compression, the frameworks of GmA tend to integrally deform along with the larger Δ rather than the straight up and down buckling of the PGA (Fig. 5c, d). This mode could optimize the deformation process globally and decrease the elastic energy to avoid its failing. It should be noted that the changed deformation mode mainly resulted from the increasing bending stiffness (D) of graphene walls. Generally, for a plate material, its buckling occurs when it is stressed beyond the yield stress, which is in direct proportion to the D of materials[20]. According to the free energy density equation established by the GmA model (Supplementary Fig. 22), the D is found to be proportional to the initial density of nanofiber (Fig. 5e), which mainly benefited from the combination of 1D and 2D nanostructures. The introduction of 1D nanofibers into 2D structure not only strengthened the stability of

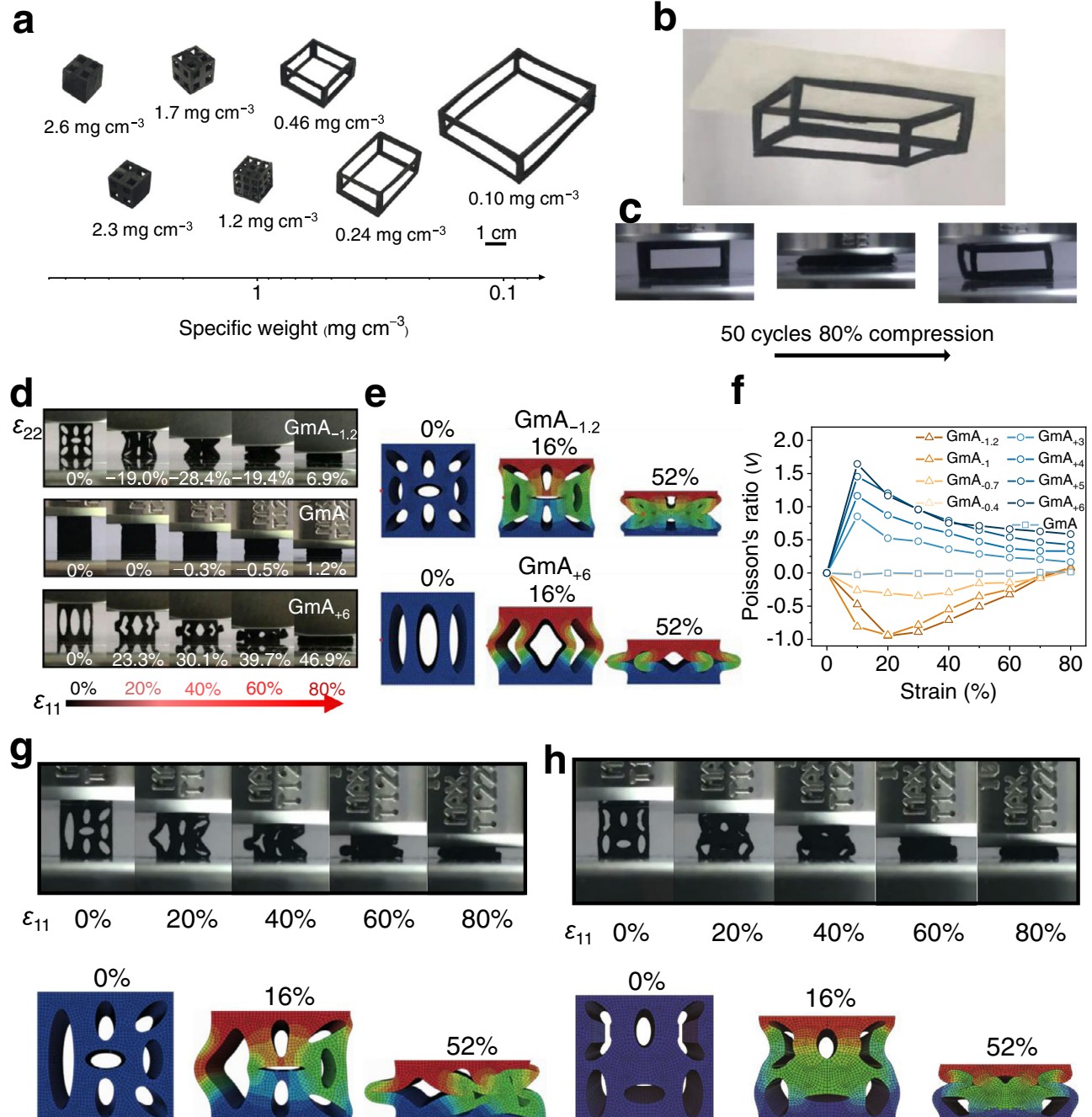

**Fig. 6 | Laser engineering meta-structures of GmAs with appealing properties.**
**a** The specific weight distributions of GmAs after laser-engraving. The pictures give specific weight and shapes of different GmAs. **b** a GmA (3.3 cm × 2.7 cm × 1 cm) with the ultralight specific weight of 0.1 mg cm$^{-3}$ can overcome the gravity relying on the electrostatic force. **c** Photographs of the GmA (3.3 cm × 2.7 cm × 1 cm) with specific weight of 0.1 mg cm$^{-3}$ before and after 50 compression cycles at 80% strain. **d** The snapshots of GmAs (0.8 cm × 0.5 cm × 0.8 cm) with different configurations during uniaxial compression, The snapshots of GmAs (0.8 cm × 0.5 cm × 0.8 cm) with different configurations during uniaxial compression, concave-shaped configuration (GmA$_{-1.2}$) (upper row), pristine configuration (GmA) (middle row), and convex-shaped configuration (GmA$_{+6}$) (lower row). **e** The finite element calculation shows the compression process of samples of concave-shaped (GmA$_{-1.2}$) and convex-shaped (GmA$_{+6}$) configurations, and the color reflects the distribution of displacement magnitude. **f** The Poisson's ratio variations of GmA$_{\pm n}$ during uniaxial compression. ± represents the positive or negative Poisson's ratio behavior of GmA. n means the n mm of major axis of the ellipses in the GmA, representing the changing hole size. **g** The snapshots of cross-section views of the GmA (0.8 cm × 0.5 cm × 0.8 cm) of the left and right asymmetry configuration and the finite element simulation process during compression process. **h** The snapshots of cross-section views of GmA (0.8 cm × 0.5 cm × 0.8 cm) of the upper and lower asymmetry configuration and the finite element simulation process during compression process. Source data are provided as a Source Data file.

every single layer, but also bound all the layer units into a whole to increase effective thickness. Thus, a rapid bending stiffness enhancement for local graphene skeleton with addition of nanofibers was demonstrated, leading to the more stable bulk deformation mode of the GmA.

## Meta-structure design of GmAs

Such characteristics of GmAs as good elastic, robust, and stiff properties pave the way to build up the unique meta-structures through laser-engraving technique. Laser-engraving can endow GmA with ultralight properties without sacrificing its high elasticity. Figure 6a

shows the GmA with configurations from the lattice-like patterns to the completely hollow frames while remaining the structure integration. The specific weight (the ratio of the mass to the space taken by the frame of GmA) can decrease to a record low value of $0.1\,\mathrm{mg\,cm^{-3}}$, and it can overcome the gravity and firmly attach to the printing paper by electrostatic force (Fig. 6b and Supplementary Movie 1). Even more, such an ultralight architecture can fully-spring back to its original height under 80% compression after 50 cycles (Fig. 6c and Supplementary Movie 2), demonstrating the lowest specific weight of GAs along with presenting the excellent fatigue resistance over the light GAs so far (Supplementary Table 1)[8,9,11,14,23–26,34–43].

Structural materials with different Poisson's ratio ($v$) are of great influence on their mechanical behaviors that drive different motions, deformations, stresses and mechanical energy variations and promise applications in various fields[7,44,45]. However, it still lacks the suitable method to effectively adjust $v$ for GAs. Pristine GA is a near-zero $v$ material owing to its lamella architectures (Supplementary Movie 3), which almost never causes the transverse deformation ($\varepsilon_{22}$) under longitude applied strain ($\varepsilon_{11}$), and the $v$ is calculated by $v = -\varepsilon_{22}/\varepsilon_{11}$. Beyond the ultralight yet robust skeleton, GmA meta-structures by laser-engraving realize the ultrawide $v$ range. The anisotropic hole patterns on the GmA engrave with the concave or convex shapes, which form the inward or outward protrusion frameworks. When compressed, the longitude stress is divided into transverse stress along the skeleton structure, causing the skeleton to bend inward or outward (Fig. 6d and Supplementary Movies 4 and 5). By changing the size of the holes, the concave or convex effects on GmA are enhanced or weaken accordingly, which will control $v$ behaviors by a large margin from negative to positive regions (Supplementary Table 2). Finite element calculation further confirms the deformation processes consistent with the corresponding multi-hole samples (Fig. 6e and Supplementary Fig. 23–25, simulation details in Supplementary note 3). Thus, an ultrawide $v$ peak value range of $-0.95 < v_{peak} < 1.64$ is achieved (Fig. 6f). To the best of our knowledge, this is also the widest $v$ range for GAs reported so far (Supplementary Table 3)[3,10,25,26,40,41,45–47]. In addition, after rational integration of the concave-shaped and convex-shaped configurations in one GmA block, more diverse and complex deformation under compression can be prepared (Fig. 6g, h), such as the left and right asymmetry, upper and lower asymmetry, largely expanding unique properties with arbitrary designability of the GmA under laser manufacturing. On the basis of the rich structures of GmAs, more applications such as actuators, strain sensors, and protection are expected (Supplementary Fig. 26).

Additionally, we can further incorporate the magnetic nanoparticles into the GmAs to fabricate various magnetically responsive actuator with high shape fidelity (Supplementary Fig. 27). Such a graphene column array (Supplementary Fig. 27a) exhibits a flexible wave-like deformation along with the movement of magnetic field (Supplementary Movie 6) and a graphene stair spring (Supplementary Fig. 27b) can carry the lightweight objects step by step under the alternating magnetic field (Supplementary Movie 7). In addition, the highly stable GmA provides a promising thermal barrier and protects a paper boat upon exposure to flame (Supplementary Fig. 28 and Supplementary Movie 8). After open-flame test, the PI nanofibers are still retained with ordered structure (Supplementary Fig. 29). GmA platforms can also be used as the templates to precisely prepare the ceramic aerogels with different shape (e.g., anatase $TiO_2$ JCPDS 21-1272 and baddeleyite $ZrO_2$ JCPDS 37-1484) (Supplementary Fig. 30). After removing of the templates, the ceramic aerogels remained the controllable inner patterns and good mechanical performances (Supplementary Fig. 30), providing key opportunities for thermal and energy management where space is limited such as in battery, wearable devices, electronics and micro-electromechanical systems.

## Discussion
Combining with the effective strengthened structures of 1D nanofibers reinforced 2D sheets, a laser-engraving strategy was proposed to construct GmAs with unique characters and rich functionalities, which opens up a super easy way to precisely regulate the GA structures from tailorable macro to well-ordered micro configurations. Specifically, the internal 1D nanofibers reinforced 2D frameworks ensure a stable bulk deformation process, leading to a significant improvement of its elasticity, robustness, and stiffness. Accordingly, laser-engraving achieves arbitrary structures of GmAs with record characteristics, such as the super stretchability (5400% reversible tensile strain), ultralight specific weight ($0.1\,\mathrm{mg\,cm^{-3}}$), and ultrawide Poisson's ratio range ($-0.95 < v_{peak} < 1.64$). Moreover, the easy procedure can incorporate polymer and particles into the ordered structure enabling the yield of objects with pre-defined geometry and functions. With these impressive properties of GmAs, this work sheds light on future developing of the multi-functional aerogels with tunable meta-structure.

## Methods
### Synthesis of GO
GOs were prepared by a water-enhanced oxidation method under low temperature of $0–5\,^{\circ}C$[48]. The basic physical and chemical structure characterizations of GO sheets are listed in Supplementary Figs. 4, 31, 32.

### Synthesis of PI nanofibers
PI nanofibers were fabricated by an electrospinning strategy, which consists of the precursor fabrication, electrospinning, and thermal imidization. The precursor is poly(amic acid) (PAA), which is prepared by condensation polymerization reaction of dianhydride and diamine. Specifically, 2 g 4,4'-oxybisbenzenamine and 30 mL N,N-Dimethylacetamide were added into a 250 mL round-bottom flask with mechanical agitation at 250 rpm to dissolve the diamine and form a transparent solution. Then setting the round-bottom flask in a cold bath to maintain the temperature below 5 °C, 2.25 g 1,2,4,5-Benzenetetracarboxylic anhydride was added to the solution in small portions within 1 h. The reaction was then stirred for 12 h, and a viscous PAA solution was prepared. The electrospinning was conducted on a HD-2335 electrostatic spinning machine (Yongkang Co. Ltd, China). The light yellow PAA solution was added into a 5 mL disposable syringe with a 22-G needle. The feed rate was set as $1.5\,\mathrm{mL\,h^{-1}}$. A plane covered Al foil as the collector to collect the PAA nanofibers, and the distance between the collector and needle is 15 cm. The applied positive voltage is 12 kV and the applied negative voltage is −2 kV. After electrospinning, a milk white PAA mat was prepared (Supplementary Fig. 2). The thermal imidization process was performed on a programmed heating tube furnace (KJ Group, China). Specifically, the PAA mat was put into a ceramic boat setting at the middle of the tube furnace. The temperature was gradually increased in the following sequence: 100, 150, 200, 250, and 300 °C, and each temperature holds 0.5 h. The heating rate is $5\,^{\circ}C\,\mathrm{min^{-1}}$. Finally, the yellow PI mats with nanofibers structure were prepared (Supplementary Figs. 2, 3).

### Synthesis of GmA
The as-prepared PI mats were cut into small pieces with an area below $0.25\,\mathrm{cm^2}$. Then PI pieces and dioxane were added into a beaker, and subsequently with a high-speed shearing treatment over 2 h by a D-130 homogenizer (Wiggens, German), a homogeneous PI dispersion was prepared. Noted that the dioxane can be moderately added during the shearing process to avoid the aggregation of the PI nanofibers. For preparing the GmA, the $10\,\mathrm{mg\,mL^{-1}}$ GO dispersion was mixed with $15\,\mathrm{mg\,mL^{-1}}$ PI dispersion with a continuous agitation, and the amount can be tuned according to the weight ratio of PI to GO. Here, three different weight ratio was prepared including 1:10, 3:10, and 5:10,

which were nominated as 0.1-GmA, 0.3-GmA, and 0.5-GmA, respectively. After mixing uniformly, the mixture was added into a home-made mold to conduct the directional freezing. The mold was a hollow cube (2.0 cm × 2.0 cm × 2.0 cm) with two open opposite face. Before using one open face was sealed by 2 mm thick polydimethylsiloxane (PDMS) (Supplementary Fig. 33). Then, the mold was put on a metal block with PDMS face contacting on the metal. Meanwhile, liquid nitrogen as the cold source to cool down the metal block to direct control ice growth. Following the freeze-drying and thermal annealing at 500 °C for an hour at Ar, the GmA was prepared (Supplementary Fig. 34). The GmAs with different graphene wall thicknesses were prepared according to the addition amount of GO and PI. Specifically, the amount of graphene oxide is fixed at 5 mg mL$^{-1}$, and the amount of PI nanofibers is increased in a series of 0.5, 1.5, and 2.5 mg mL$^{-1}$, respectively. Thus, the wall thickness would gradually change. The laser-engraving GmA was manufactured by a LSU3EA UV laser machine (HGTECH, China) with a 355 nm laser beam at a spot size of 10 μm. The laser power was controlled to be 3 ± 0.1 W with a pulse repetition frequency of 20 kHz, and a marking velocity of 1000 mm s$^{-1}$. The patterns were prepared in advance by software. The laser-engraving process is a rapid localized heating and combustion of graphene and PI. The gases generated from combustion would tear graphene layer. Residues from insufficient combustion remain on the edge of graphene. The related process was shown in Supplementary Figs. 35 and 36.

### Synthesis of magnetically responsive GmA and ceramic aerogels
For the preparation of magnetically responsive GmA, Fe$_3$O$_4$ nanoparticles (25 wt% in water, 10-30 nm, Supplementary Fig. 37) of equal mass to GO were added into the GO and PI mixture before the directional freeze. The following procedure was similar with GmA preparation. For preparation of ceramic aerogels. the GmA was immersed in the aqueous solution which contains 0.3 M Ti(SO$_4$)$_2$ (Zr(SO$_4$)$_2$ for ZrO$_2$-GmA) and 1.8 M urea. Assisted by the vacuum system, the gas was excluded from the GmA. And the immersed GmAs were put in 90 °C water bath for 3 h, followed by freeze-drying and an 800 °C annealing treatment at air, the TiO$_2$-GmA was prepared. The density of TiO$_2$-GmA and ZrO$_2$-GmA was 46 mg cm$^{-3}$ and 36 mg cm$^{-3}$, respectively.

### Characterizations
Scanning electron micrographs (SEM) were performed by using a Field Emission Zeiss Gemini300 with an accelerating voltage range from 5.0 kV to 10.0 kV (Zeiss, German). Transmission electron microscopy (TEM) images were taken by the Tecnai F20 transmission electron microscope (FEI Corporation, USA), which is operated at 200 kV. XPS spectra were conducted on an Escalab 250 photoelectron spectrometer (Thermo Fisher Scientific, USA). The Raman spectra of GO and rGO were studied by means of a LabRAM HR Evolution with a 532 nm excitation laser (HORIBA Jobin Yvon, France). The photos were taken by an iPhone 7 mobile phone (Apple Inc., USA). Attenuated total reflection Fourier transform infrared (ATR-FTIR) spectra were recorded on an UATR Two FT-IR spectrometer (Perkin Elmer, USA). X-ray diffraction (XRD) patterns were collected by the use of a D8 Advance X-ray diffractometer with Cu Kα radiation (λ = 0.15418 nm, Bruker, Germany) at scanning rate of 5 degree min$^{-1}$. Thermogravimetric analysis (TGA) was characterized a TA-Q50 TGA (Netzsch, Germany) under a heating rate of 10 K min$^{-1}$, the protect gas include air and N$_2$. Dynamical mechanical analysis (DMA) were conducted a Q800 mechanical analyzer (TA instruments, USA). Mechanical tests were conducted on an Instron 3342 universal testing machine (Instron, USA) equipped with a 100 N load cell. Specifically, the tensile test was performed with a constant loading rate of 1 mm min$^{-1}$ and a gauge length of 10 mm. For the compressive test, the samples were shaped into cubic appearance (8 mm × 8 mm × 8 mm). The compressive rate was set 8 mm min$^{-1}$.

## Data availability
The data generated in this study are provided in the Supplementary Information and Source Data file. Source data are provided with this paper.

## Code availability
The code supporting this study is available from the corresponding author upon request.

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

## Acknowledgements

L.Q. acknowledges the financial support from the National Science Foundation of China (Nos. 52073159, 22035005, 21774015, 52022051, 22075165, 52090030), NSFC-MAECI (51861135202), NSFC-STINT (21911530143), State Key Laboratory of Tribology (SKLT2021B03), and Tsinghua-Foshan Innovation Special Fund (2021THFS0501). M.W. is grateful for financial support from the National Science Foundation of China (No. 22105040), the Fujian Science & Technology Innovation Laboratory for Optoelectronic Information of China (2021ZZ127), Natural Science Foundation of Fujian Province of China (2021J01588), and Fuzhou University Testing Fund of precious apparatus (2022T005). F.L. also acknowledges the support from the National Science Foundation of China (11972349, 11790292) and the Strategic Priority Research Program of the Chinese Academy of Sciences (Grant No. XDB22040503).

## Author contributions

L.Q. and M.W. designed the experiments. M.W. performed the experiments. F.L. conducted the computational studies. Y.W. assisted in the design of figures. H.G., Y.H., H.M., C.Y., H.W.C., H.H.C., C.L., and L.J. participated and gave advice on experiments. L.Q. supervised the entire project. All authors discussed the results and reviewed the manuscript.

## Competing interests

The authors declare no competing interests.
