## [Peer Review File · Nature Communications]

REVIEWER COMMENTS

Reviewer #1 (Remarks to the Author):

This paper reports the application of laser-engraving technique for shaping of graphene aerogel into a variety of structures. Graphene aerogel has been intensively investigated in the past decade for its wide potential applications. The attempt to fabricate graphene aerogels with a wider spectrum of structure would be helpful for the researchers in this field. The manuscript itself is highly contented and clearly presented. However, I would suggest the authors to further improve their manuscript to merit publication in Nature Communications, by focusing both on the mechanism of the laser-cutting technique and the properties of the resulted materials. Some specific suggestions:

1. Regard to the mechanism of the laser-cutting fabrication of the graphene-PI composite aerogel, what is the exact process? What was the temperature during the process? What are the chemical reactions? Are PI nanofibers burned away? What does the surface evolve at the 'cutting' surface? Are there a dense layer at the 'cutting' surface?
2. What is the resolution of the laser cutted structure? And how this is related to the properties of the resulted material. As this is crucial for the future application of this technique in making graphene aerogels. For sure, this has been investigated in traditional materials, but is unclear when used for graphene aerogel, as in the current research.
3. The authors have included several mechanical test trying to demonstrate the properties of their materials. However, I found it difficult to capture the advantages. I would suggest the authors provide more Ashby charts for better comparison.
4. I appreciate the authors realized a wide range of negative Poisson's ratio by introducing additional porosity by laser cutting. I would suggest the authors further demonstrate or at least discuss the potential applications of this kind of property, particularly when this material has an extremely low density. I would guess it is no longer suitable for applications related to energy absorption.
5. Properties in Figure 7 is a bit redundant. The necessity of the magnetic graphene aerogel and composite graphene-ceramic aerogel requires additional justification.

Reviewer #2 (Remarks to the Author):

Here, the authors reported a superelastic graphene meta-aerogel that is composed of the nanofibers with graphene skeleton at the microstructure, and the laser-engraving technique further finely controls the macro morphology of graphene meta-aerogel. The enhancement mechanism of the microstructure is discussed and verified step by step in this manuscript by the experimental and theoretical results. Rich structure design and record performances presented are of interest to the graphene community—integrating the good material processing and various functionalities in the graphene aerogel appears to be important. This study is well organized. For these reasons, I believe that this work shall attract wide attention and therefore the manuscript can be accepted for publication in Nature Communications after minor revision.

1. The scale bar is suggested to be added in Fig. 1 (d).
2. What are the advantages of the electrospun PI nanofibers when compared to the other nanofibers?
3. In Equation (1) the statistics are made every five degrees in angle. Is there any reason? In other words, whether the angle interval will affect the statistical results.
4. In Fig.5 (b), a comparison of relative strain energies between calculations and experiments is shown. As stress and strain curves could also be obtained from calculations and experiments, why not compare them, which could be a more direct comparison from my viewpoint.
5. A series of GmAs with different graphene wall thicknesses were prepared. How to control the thickness change?
6. In Fig.6 (h), FEM results are shown in an opposite direction comparing to the experiments, is there any purpose?
7. The thermal stability of GmA should be evaluated in air. The TGA measurements are suggested.
8. The density of ceramic aerogels should be given.
9. There are a few typos and errors in the current manuscript that need to be carefully corrected.

Reviewer #3 (Remarks to the Author):

Comments on Superelastic graphene meta-aerogel "sculptures"
Authors; Wu et al.

This communication demonstrates a laser-engraving strategy toward graphene meta-aerogels (GmAs) with unusual characters. Authors state that the nanofiber-reinforced networks convert the graphene walls' deformation from the plastic buckling to the bulk deformation during the compression process, ensuring the highly elastic, robust, and stiff nature. The work is interesting, thus reviewer thinks that the paper may be considered for potential publication in NC provided authors can properly address following questions:

1. Please explain how to from plastic deformation convert to elastic deformation? This is not correct.
2. The captions in Fig. 1-3 are not suitable. Some pictures are not graphene meta-aerogels (GmAs).
3. In Supplementary Fig. 21. Stress-strain curves of GA with different density of fibers, why authors select Eq. 2 free energy density form? The mechanical properties of graphene meta-aerogel should be high nonlinear, for so large deformation, authors use bending stiffness D formula is not correct.

Point-by-Point Responses to Reviewers' Comments

Reviewer #1 (Remarks to the Author):

This paper reports the application of laser-engraving technique for shaping of graphene aerogel into a variety of structures. Graphene aerogel has been intensively investigated in the past decade for its wide potential applications. The attempt to fabricate graphene aerogels with a wider spectrum of structure would be helpful for the researchers in this field. The manuscript itself is highly contented and clearly presented. However, I would suggest the authors to further improve their manuscript to merit publication in Nature Communications, by focusing both on the mechanism of the laser-cutting technique and the properties of the resulted materials. Some specific suggestions:

Reply: We thank the referee for the positive comments and constructive suggestions for improving our manuscript. We have conducted additional experiments and added more discussions in the revised manuscript. Some modifications and explanations are summarized below.

1. Regard to the mechanism of the laser-cutting fabrication of the graphene-PI composite aerogel, what is the exact process? What was the temperature during the process? What are the chemical reactions? Are PI nanofibers burned away? What does the surface evolve at the 'cutting' surface? Are there a dense layer at the 'cutting' surface?

Reply: Thanks for your questions. In general, laser processing involves rapid energy input and energy deposition on a solid's surface. The pulsed/continuous laser energy is absorbed by the electrons in this process. Then the electrons interact with the lattice to complete the energy transfer (10^{-11} – 10^{-12} s). And the thermal equilibrium is established between the lattices with increasing temperature and kinetic energy of the lattice. The resultant temperature depends on the absorbed energy, thermal diffusion rate of materials, and work conditions during laser irradiation. In our experiment, the average laser power is 3 ± 0.1 W with a pulse repetition frequency of 20 kHz. The ablation threshold of graphene materials is about 0.16 – 0.21 J cm⁻² (Appl. Surf. Sci. 2013, 276, 133; Appl. Phys. A, 2012, 109, 291.). Thus, the input energy of a 3 W laser

can easily break the graphene layer.

Fig. R1. The photograph of the thermocouple and probe.

It is difficult to precisely measure the accumulated temperature of graphene sheets. To evaluate the temperature during the laser process, a thermocouple with a metal probe was used (Fig. R1). The detection area of the probe is 1 mm^2 . The laser spot directly irradiates the probe to record the temperature. As shown in Fig. R2, the measured temperature increased sharply to $1300 \text{ }^\circ\text{C}$ within the first few seconds, confirming the rapid energy input of laser. Subsequently with 60 s of continuous irradiation, the temperature gradually stabilized at around $1000 \text{ }^\circ\text{C}$. Furthermore, it should be noted the graphene is a better light absorbing material compared to conventional metallic materials. The instantaneous temperature of graphene materials might be higher under laser irradiation.

Fig. R2. The temperature measurements under laser irradiation. **a** The temperature variation under spot laser irradiation. **b** After 10 s laser irradiation. **c** After 45 s laser irradiation.

After understanding the instantaneous temperature under laser irradiation, the

possible chemical reactions of the graphene-PI composite aerogel could be further deduced. In general, as the temperature exceeds 500 °C, the graphene and PI will be rapidly oxidized by oxygen. To confirm possible products, the thermogravimetric analysis/mass spectrometry (TGA-MS) measurement was performed. The heating rate is 10 °C min⁻¹. Fig. R3 displays TGA-MS curves of the PI and GmA in the air condition. At a temperature below 500 °C, the PI and GmA show almost no weight loss. As the temperature rises above 500 °C, the PI and GmA begin to decompose with a rapid weight loss. During this process, the main products are CO₂ (m/z:44), CO (m/z:28), H₂O (m/z:18), and a small amount of nitride. Correspondingly, in the laser irradiation process with an instantaneous temperature surpassing 1000 °C, oxidation reactions of carbon and polymer materials are the most likely to occur, leading to a rapid localized combustion of graphene and PI. But it should be noted that the laser spot will have an enlargement during the irradiation. The temperature at the edge of the laser may be lower than that at the spot center, resulting in less oxidation. As shown in Fig. R4a-c, an electrospun PI film was directly cut by laser. The width of the laser cut is about 200 μm (Fig. R4a). However, at the edge of the PI film incision, the electrospun PI nanofibers have been melted into balls, which was derived from the enlargement of laser spot. At the edge of laser spot, the PI nanofibers cannot conduct the heat away, leading to an insufficient combustion and carbonization. The EDS line scanning also confirmed the spherical PI exhibits more carbon and oxide elements than those of PI nanofibers (Fig. R4d,e).

Fig. R3. TGA-MS spectrum of PI and GmA.

Fig. R4. SEM images of electrospun PI film cut by laser. **a** Scale bar is 200 μm . **b** Scale bar is 50 μm . **c** Scale bar is 20 μm . **d** SEM image of EDS line scanning. **e** EDS line scanning.

As the PI nanofibers are incorporated with graphene, the surface evolution of the GmA under laser-engraving will be different. Because of the good thermal conductivity of graphene, the heat cannot build up to melt the PI nanofibers into balls (Fig. R5a, c). However, the rapid combustion of graphene and PI will produce fast-flowing airflow that can break the graphene walls and result in partially cracked structures (Marked in red circles in Fig. R5a). In contrast, the GmA surface ripped by hands exhibits a relatively complete structure without obvious fractures (Fig. R5b, d). Furthermore, the effect of laser-induced combustion could be observed at the edge of graphene walls (Fig. R5e, g). There are residual flocculent structures, which are formed by the insufficient combustion of the carbon skeleton. In contrast, the tearing edges of graphene walls show a smooth structure (Fig. R5f, h). Thus, the ‘cutting’ surface usually suffers from insufficient combustion. Its degree of oxidation is lightly higher than non-laser-cutting surfaces. XPS spectrum display the O/C ratio of the laser-cutting surface and non-laser-cutting surfaces are 0.104 and 0.095, respectively. Meanwhile, the inadequate combustion remnants also remain at the ‘cutting’ surface, which are typically loose flocc-like structures, instead of a dense layer.

Fig. R5. SEM images of GmA surfaces after laser-cutting and tearing. **a, c, e, g** GmA surfaces after laser-cutting under different magnification. **b, d, f, h** GmA surfaces after tearing under different magnification.

In summary, in an exact process of laser-engraving, laser irradiation will cause rapid localized heating of graphene and PI composite aerogels, leading to the combustion of carbon and polymer materials. Meanwhile, the gases generated from combustion will tear graphene layer. At the edges of the laser spot, the temperature is insufficient for full combustion, which often leaves some partially oxidized products behind. It is the remained ‘cutting’ surface with some floc-like structures.

Related discussion and figures have been added in revised manuscript and Supplementary Information. (Please see page 23-24, line 440-443 and revised

Supplementary Information, page 42-43, line 417-452.)

2. What is the resolution of the laser cutted structure? And how this is related to the properties of the resulted material. As this is crucial for the future application of this technique in making graphene aerogels. For sure, this has been investigated in traditional materials, but is unclear when used for graphene aerogel, as in the current research.

Reply: Thanks for your insightful comments. In this study, the laser was carried out on a nanosecond laser machine with a laser spot size of 10 μm and a pulse width of 14 ns. However, in practice, the fast energy input will make the spot enlargement and have a wider engraving area.

To test the resolution of the laser-engraving structure, a series of laser spots with different sizes were carried out on the GmA, including 1 mm, 600 μm , 200 μm , 100 μm , and 20 μm diameters. As shown in Fig. R6, all GmAs have a spherical ablation hole after laser irradiation. The holes were measured to be 1.02 mm, 751 μm , 365 μm , 218 μm , and 216 μm in diameter, respectively. By comparison, only the 1 mm diameter achieves the default dimensions. The laser-engraving GmAs with spot diameters of 600 and 200 μm show a 150 μm enlargement. Additionally, as the laser spot diameters decrease to 100 μm and 20 μm , the hole sizes become nearly identical, and they are both 200 μm . As a result, we can conclude (1) the smallest laser-engraving size that this nanosecond laser machine can achieve is 200 μm , even if the laser spot is set to a smaller value; (2) this laser machine can accurately perform the engraving with a resolution requirement larger than 1 mm; (3) the laser beam will enlarge to some extent, as the engraving size is between 1 mm and 200 μm .

In this study, the laser-engraving technique is mainly used to shape the macro-morphology of graphene aerogels, such as GmAs with flower structures (Fig. 1c), GmAs with lattice-like patterns (Fig. 6a), and GmAs with multi-hole patterns (Fig. 6d, g, h). The processing sizes are typically higher than a few millimeters, which the

engraving resolution of the nanosecond laser can well match. Furthermore, as compared to the shaping methods assisted by frozen molds and the 3D printing technique, laser-engraving can realize macro-morphology processing in a few seconds. The quick and accurate process also confirms the advantage of laser-engraving.

Fig. R6 SEM images of GmA after different laser spot engraving. The diameter of laser spot is **a** 1 mm, **b** 600 μm , **c** 200 μm , **d** 100 μm , and **e** 20 μm , respectively.

Related discussion have been added in revised manuscript and Supplementary Information. (Please see page 23-24, line 440-443 and revised Supplementary Information, page 42, line 434-438.)

3. The authors have included several mechanical test trying to demonstrate the properties of their materials. However, I found it difficult to capture the advantages. I would suggest the authors provide more Ashby charts for better comparison.

Reply: Thanks for your constructive comments. “Ashby” or “bubble” charts are powerful tools for helping in materials selection and evaluation. Per your suggestion, we have included Ashby charts in our revised manuscript, which depict the strength or modulus versus density for various types of materials (Fig. R7).

As shown in the Ashby plots, aerogels are ultralight materials with a density typically less than 10 mg cm^{-3} . The strength and modulus of aerogels are mainly located

in the lower left area of the figures. Among aerogels, GmAs are typical materials with good strength and modulus. Considering the excellent resilience (Fig. R7c), the GmA is hopeful for further applications with good durability and processability.

Related discussion and the Fig. R7 have been added in revised manuscript and Supplementary Information. (Please see page 12, line 201-205 and revised Supplementary Information, page 19, line 213-224.)

Fig. R7. Ashby plots for various types of materials. **a** Strength vs. density. The strength of GmA is the stress under 80% compression strain. **b** Modulus vs. density. The data was collected from references and CES EduPack 2019, ANSYS Granta © 2020 Granta Design. **c** Stress retention versus density of carbon aerogels under 50% compression strain.

4. I appreciate the authors realized a wide range of negative Poisson's ratio by introducing additional porosity by laser cutting. I would suggest the authors further demonstrate or at least discuss the potential applications of this kind of property, particularly when this material has an extremely low density. I would guess it is no longer suitable for applications related to energy absorption.

Reply: Thanks for your valuable comments! Compared with common materials,

negative Poisson's ratio materials have superior shear resistance, indentation resistance, and fracture toughness, which makes them suitable for energy absorption applications, such as aerospace, defense, and sports protection. In our study, laser-engraving can achieve a wide range of Poisson's ratios by designing different hole structures on GmAs. As the size of the holes is small, the density of GmA does not change significantly. Compared with the original GmA, the density of GmAs_n with different hole size decreased by 31%, 22%, 11%, and 3.5% according to the calculation of the reduced volume by the holes. They still have energy absorption capability and can be utilized as cushioning materials. Additionally, the density of original GmA can be adjusted by adding graphene and PI, which may allow the density to meet the requirements even if the weight of GmA is reduced by the holes.

On the other hand, for the GmA with low density, although it can no longer guarantee its application in mechanical energy absorption, light skeletons will be more sensitive to stress and strain. Recent articles reported the auxetic structure is a universal strategy for enhancing piezoresistive sensitivity (Matter 2022, 5, 1-16; Adv. Mater., 2018, 30, 1706589.). In our study, the ultra-light GmA could be also applied in small strain sensors, which is hopeful to be more sensitive because of its lighter and more flexible skeleton structures. Fig. R8 shows under the condition of small strain, the GmA with negative Poisson's ratio, such as GmA_{-0.7} and GmA_{-1.2}, exhibits larger gauge factor (GF) compared with the GmA without any holes, demonstrating the better sensitivity. The GF is calculated by $(\Delta R/R_0)/\varepsilon$, where ΔR is the resistance change under compression, R_0 is the resistance before straining, and ε is the applied strain. Additionally, the ultra-light GmA can be used as a platform to incorporate other particles to achieve more functionalities. In our study, the magnetically responsive actuator has been demonstrated by the low density GmA. Other materials such as temperature-responsive polymers and shape memory polymers can also be added in GmAs. These responsive functions can be retained even at ultra-light densities.

Fig. R8. Normalized resistance $\Delta R/R_0$ values at a compression strain of 20%. **a** GmA without holes. **b** GmA_{-0.7}. **c** GmA_{-1.2}. ΔR is the changing value of the resistance during compression. R_0 is original resistance without compression.

Besides, another important advantage of laser-engraving is that materials can be arbitrarily designed. In addition to structures with a negative Poisson's ratio, laser-engraving can endow GmA with anisotropy, which can be employed in building materials for anisotropic transfer and absorption of sound or heat (Fig. R9).

Fig. R9. Anisotropic GmA for sound/heat transfer and absorption.

Related discussion and the Fig. R8 have been added in revised manuscript and Supplementary Information. (Please see page 20, line 356-358 and revised Supplementary Information, page 32, line 317-333.)

5. Properties in Figure 7 is a bit redundant. The necessity of the magnetic graphene aerogel and composite graphene-ceramic aerogel requires additional justification.

Reply: Thanks for your thoughtful suggestions! Fig. 7 illustrates the multifunctionalities of GmAs in the original manuscript, which discusses the use of GmA as a matrix to incorporate magnetic nanoparticles and ceramic precursors, further

endowing the GmA with magnetically actuation and ceramization capabilities. These can also be considered as a complement to the good mechanical properties of GmA and laser-engraving. We agree with you that the Fig. 7 is a bit redundant. The Fig. 7 has been added in Supplementary Information and the corresponding discussion of Fig. 7 has been simplified. **(Please see page 20-21, line 361-367 and revised Supplementary Information, page 33-35, line 335-369.)**

Reviewer #2 (Remarks to the Author):

Here, the authors reported a superelastic graphene meta-aerogel that is composed of the nanofibers with graphene skeleton at the microstructure, and the laser-engraving technique further finely controls the macro morphology of graphene meta-aerogel. The enhancement mechanism of the microstructure is discussed and verified step by step in this manuscript by the experimental and theoretical results. Rich structure design and record performances presented are of interest to the graphene community—integrating the good material processing and various functionalities in the graphene aerogel appears to be important. This study is well organized. For these reasons, I believe that this work shall attract wide attention and therefore the manuscript can be accepted for publication in Nature Communications after minor revision.

Reply: We are grateful for your positive recommendation. According to your suggestions, we have carried out additional experiments and discussions in the revised manuscript. We also have carefully revised figures and main text. Some modifications and explanations are summarized below.

1. The scale bar is suggested to be added in Fig. 1 (d).

Reply: Thanks for your comments. We have added the scale bar in Fig.1c, d. (Please see page 6, Fig. 1)

2. What are the advantages of the electrospun PI nanofibers when compared to the other nanofibers?

Reply: Thanks for your good points. The electrospun PI nanofibers used in this work have three unique advantages which are far better than other nanofibers. First of all, as a network skeleton for enhancing graphene, the nanofibers should possess excellent mechanical performances including high strength, rigidity, and good flexibility. PI nanofibers can well content these requirements. PI is composed of rigid heterocyclic imide rings and aromatic benzene rings, ensuring its strong and rigid macromolecular backbones. The tensile strength and modulus of electrospun PI nanofibers are 300-1000 MPa and 5-15 GPa (Prog. Polym. Sci. 2016, 61, 67.), which are better than other nanofibers such as the common cellulose nanofibers. Moreover, the aromatic structure of PI can form a good interaction with the carbon framework of graphene through the π - π interaction (Fig 2c), providing a tight contact between graphene sheets and PI nanofibers. Secondly, the PI nanofibers are thermally stable. Most polymer nanofibers will gradually decompose when the temperature is higher than 300 °C, which cannot meet the reduction temperature of graphene oxide. Because the reduction of graphene oxide often involves a high-temperature deoxidation process. While PI nanofibers can maintain stability at 500 °C (Supplementary Fig. 28b), demonstrating the unique advantage of composites of PI and graphene. Thirdly, the electrospun PI nanofibers possess an appropriate dimension with an average diameter of 207 nm, which matches well with the size of graphene layer. In the SEM observation, the PI nanofibers are firmly adhered to graphene sheets, forming a continuous monolithic structure. Thus, the PI nanofibers prepared by the electrospinning techniques are the preferred nanofiber network in this work over other nanofibers.

3. In Equation (1) the statistics are made every five degrees in angle. Is there any reason? In other words, whether the angle interval will affect the statistical results.

Reply: Thanks for your questions. As the texture characteristics have been captured, any angle interval could be used for statistics theoretically. However, limited by the number of samples, the statistical results with smaller angle becomes

nonsmoothed, indicating the “resolution” is not able to support such an accurate analysis. Both considering the “resolution” limitation (the number of samples) and the accuracy of analysis, five degrees’ angle interval is applied in our study.

4. In Fig.5 (b), a comparison of relative strain energies between calculations and experiments is shown. As stress and strain curves could also be obtained from calculations and experiments, why not compare them, which could be a more direct comparison from my viewpoint.

Reply: Thanks for your question. In our theoretical calculations, only a representative configuration is used to study the deformation characteristics of the system. Due to the size of the configuration is not big enough, the stress fluctuation is significant and thus not appropriate to compare with experimental data directly. However, strain energy is the integral of stress with strain, which has much less fluctuation than stress and is therefore used for direct comparisons.

5. A series of GmAs with different graphene wall thicknesses were prepared. How to control the thickness change?

Reply: The graphene wall thickness is regulated by controlling the amount of graphene oxide and PI nanofibers. In the Methods section, the synthesis of GmA part describes the preparation of samples with different weight ratio including 0.1-GmA, 0.3-GmA, and 0.5-GmA. In order to prepare GmAs with different graphene wall thicknesses, the amount of graphene oxide is fixed and the amount of PI nanofibers is increased. For example, for 0.1-GmA, 0.3-GmA, and 0.5-GmA, the amount of graphene oxide is 5 mg mL^{-1} , and the amount of PI nanofibers is 0.5, 1.5, and 2.5 mg mL^{-1} , respectively. Thus, after freeze-drying and thermal reduction, the wall thickness was changed.

Related preparation process of GmAs with different thicknesses has been added in revised manuscript. **(Please see page 23, line 432-436.)**

6. In Fig.6 (h), FEM results are shown in an opposite direction comparing to the experiments, is there any purpose?

Reply: Thanks for your reminder. We have now changed the direction of FEM results to facilitate their comparisons with experimental results. (Please see page 18, Fig. 6)

7. The thermal stability of GmA should be evaluated in air. The TGA measurements are suggested.

Reply: Thanks for your kind comments. The TGA measurement was conducted to evaluate the thermal stability of GmA in air (Fig. R1). The heating rate is $10\text{ }^{\circ}\text{C min}^{-1}$. The TGA curve presents three regions. (1) In the process of increasing the temperature from $50\text{ }^{\circ}\text{C}$ to $500\text{ }^{\circ}\text{C}$, the GmA has almost no weight loss (less than 1%), indicating that the reduced graphene oxide and PI nanofibers are stable below $500\text{ }^{\circ}\text{C}$. (2) As the temperature is higher than $500\text{ }^{\circ}\text{C}$, the weight loss of GmA begins again, which is derived from the oxidation reaction of reduced graphene oxide and decomposition of the polymer carbon backbone. (3) As the temperature is higher $600\text{ }^{\circ}\text{C}$, the weight of residual carbon skeletons is constant.

The TGA measurement and corresponding analysis have been added in Supplementary Information. (Please see Supplementary Information, page 34-35, line 353-363.)

Fig. R1. The thermogravimetric analysis (TGA) of GmA in air.

8. The density of ceramic aerogels should be given.

Reply: Thanks for your suggestion! The density of TiO₂-GmA and ZrO₂-GmA was 46 mg cm⁻³ and 36 mg cm⁻³, respectively. The corresponding description has been added in the revised manuscript. **(Please see page 24, line 452-453)**

9. There are a few typos and errors in the current manuscript that need to be carefully corrected.

Reply: Thank you very much for your valuable comments! We have double checked and polished the English spelling and grammar of the manuscript. The modified texts are marked in red in the revised manuscript.

Reviewer #3 (Remarks to the Author):

Comments on Superelastic graphene meta-aerogel “sculptures”

Authors; Wu et al.

This communication demonstrates a laser-engraving strategy toward graphene meta-aerogels (GmAs) with unusual characters. Authors state that the nanofiber-reinforced networks convert the graphene walls’ deformation from the plastic buckling to the bulk deformation during the compression process, ensuring the highly elastic, robust, and stiff nature. The work is interesting, thus reviewer thinks that the paper may be considered for potential publication in NC provided authors can properly address following questions:

Reply: We sincerely acknowledge your positive comments and constructive suggestions for further improving our manuscript. As suggested, we have corrected our inaccurate description and added more discussions, especially regarding theoretical simulation.

1. Please explain how to from plastic deformation convert to elastic deformation? This is not correct.

Reply: Thanks for your question. Yes, from the perspective of solid mechanics, once the macroscopic plastic deformation takes place, it can never recover to its initial state after unloading. In our study, we would like to emphasize the plastic events occur at microscopic scale in graphene meta-aerogels, which brings energy loss but only negligible residual deformation (please see in Fig. 3a, g, d and Fig. 4a-h), instead of macroscopic plastic of graphene aerogels. As these microscopic plastic events does not induce macroscopic plastic deformation, it endows graphene meta-aerogels with excellent resilience. From this perspective, this system only shows elasticity at macroscopical scale, thus there is no need to covert plastic deformation to elastic deformation for its recovery.

For clarity, we have revised the description of “plastic deformation” and changed “plastic buckling” to “microscopic buckling” for differentiating from macroscopic

plastic deformation. (Please see page 2, line 27, page 4, line 63, page 5, line 81, page 12, line 210, page 13, line 215, page 15, line 255)

2. The captions in Fig. 1-3 are not suitable. Some pictures are not graphene meta-aerogels (GmAs).

Reply: Thanks for your kind comments. We have carefully checked and revised all the Figures and captions. (Please see page 6, line 91-98,) The legend in Fig. 3b was revised from “DPGF” to “GmA”. (Please see page 10, Fig. 3b)

3. In Supplementary Fig. 21. Stress-strain curves of GA with different density of fibers, why authors select Eq. 2 free energy density form? The mechanical properties of graphene meta-aerogel should be high nonlinear, for so large deformation, authors use bending stiffness D formula is not correct.

Reply: According to << M. Rubinstein, Ralph H. Colby, - Polymer Physics (Chemistry) (2003)>>, the free energy density of assembly could be given as a virial expansion in powers of fundamental element's density, i.e.

$$F = \sum_{i=0}^N a_i \rho^i,$$

where F represents free energy density, a_i is expansion coefficient of i^{th} term, and ρ is fundamental element's density. Here, since there two types of fundamental elements in GmAs, i.e. fibers and graphene sheets, its free energy density could be written as

$$F = \sum_{i=0}^N \sum_{j=0}^N a_{ij} \rho_f^i \rho_{2D}^j$$

where a_{ij} is expansion coefficient, and ρ_f^i and ρ_{2D}^j is the density of fiber and graphene sheets, respectively. Only first few terms is enough to accurately describe stress-strain curves of GmAs with different density of fibers, and the obtained form is Eq.(2) in the Supplementary, the corresponding fitting parameters are also provided.

Yes, we agree your opinion that for such large deformation the form of bending stiffness is not correct. However, the current form of bending stiffness D is still considered as a good index that could roughly reflect the resistance to the bending deformation if the reference configuration is set close enough to current configuration

(analogue to the concept of tangent modulus). During this small deformation process (from reference to current configuration), the majority of stress is contributed by graphene sheets with small deformation, since the contribution from graphene sheets with post-buckling behavior is quite limited. As the bending stiffness of graphene sheets with small deformation could be described by Eq.(2), it is therefore used to roughly evaluate the resistance to bending deformation in an average manner, despite its physical meaning is not so transparent in this situation.

Point-by-Point Responses to Reviewers' Comments

Reviewer #1 (Remarks to the Author):

The authors have addressed all my previous concerns. I recommend its publication.

Reply: We really appreciate the referee for the positive recommendation and constructive suggestions for improving our manuscript.

Reviewer #2 (Remarks to the Author):

The authors resolved all reviewers' concerns and it seems now ready for publication.

Reply: We greatly acknowledge the reviewer for the positive recommendation and are grateful for the helpful and constructive comments to the manuscript.

Reviewer #3 (Remarks to the Author):

In revised version of manuscript, authors have properly addressed my questions and main concerns. The manuscript may be accepted as is.

Reply: We greatly acknowledge the reviewer for the positive recommendation and are grateful for the helpful and constructive comments to the manuscript.